# Magnesium Accumulation in Two Contrasting Varieties of *Lycopersicum esculentum* L. Fruits: Interaction with Calcium at Tissue Level and Implications on Quality

**DOI:** 10.3390/plants11141854

**Published:** 2022-07-15

**Authors:** Cláudia Campos Pessoa, Fernando C. Lidon, Ana Rita F. Coelho, Ana Coelho Marques, Diana Daccak, Inês Carmo Luís, João Cravidão Caleiro, José Carlos Kullberg, Paulo Legoinha, Maria Graça Brito, José Cochicho Ramalho, Maria José Silva, Ana Paula Rodrigues, Mauro Guerra, Roberta G. Leitão, Paula Scotti Campos, Isabel P. Pais, José N. Semedo, Maria Manuela Silva, Carlos Galhano, Nuno Leal, Fernando H. Reboredo, Maria Fernanda Pessoa, Manuela Simões

**Affiliations:** 1Earth Sciences Department, Faculdade de Ciências e Tecnologia, Universidade Nova de Lisboa, Campus da Caparica, 2829-516 Caparica, Portugal; fjl@fct.unl.pt (F.C.L.); arf.coelho@campus.fct.unl.pt (A.R.F.C.); amc.marques@campus.fct.unl.pt (A.C.M.); d.daccak@campus.fct.unl.pt (D.D.); idc.rodrigues@campus.fct.unl.pt (I.C.L.); jc.caleiro@campus.fct.unl.pt (J.C.C.); jck@fct.unl.pt (J.C.K.); pal@fct.unl.pt (P.L.); mgb@fct.unl.pt (M.G.B.); acag@fct.unl.pt (C.G.); n.leal@fct.unl.pt (N.L.); fhr@fct.unl.pt (F.H.R.); mfgp@fct.unl.pt (M.F.P.); mmsr@fct.unl.pt (M.S.); 2GeoBioTec Research Center, Faculdade de Ciências e Tecnologia, Universidade Nova de Lisboa, Campus da Caparica, 2829-516 Caparica, Portugal; cochichor@mail.telepac.pt (J.C.R.); mjsilva@isa.ulisboa.pt (M.J.S.); paula.scotti@iniav.pt (P.S.C.); isabel.pais@iniav.pt (I.P.P.); jose.semedo@iniav.pt (J.N.S.); abreusilva.manuela@gmail.com (M.M.S.); 3PlantStress & Biodiversity Lab, Centro de Estudos Florestais (CEF), Instituto Superior de Agronomia (ISA), Universidade de Lisboa (ULisboa), Quinta do Marquês, Avenida da República, 2784-505 Oeiras, Portugal; 4PlantStress & Biodiversity Lab, Centro de Estudos Florestais (CEF), Instituto Superior de Agronomia (ISA), Universidade de Lisboa (ULisboa), Tapada da Ajuda 1349-017 Lisboa, Portugal; anadr@isa.ulisboa.pt; 5LIBPhys-UNL, Physics Department, Faculdade de Ciências e Tecnologia, Universidade Nova de Lisboa, Campus da Caparica, 2829-516 Caparica, Portugal; mguerra@fct.unl.pt (M.G.); rg.leitao@fct.unl.pt (R.G.L.); 6Instituto Nacional de Investigação Agrária e Veterinária I.P. (INIAV), Quinta do Marquês, Avenida da República, 2780-157 Oeiras, Portugal; 7ESEAG-COFAC, Avenida do Campo Grande 376, 1749-024 Lisboa, Portugal

**Keywords:** leaf gas exchange, *Lycopersicum esculentum* L., magnesium accumulation, physicochemical parameters, tomato varieties

## Abstract

As the productivity and quality of tomato fruits are responsive to Mg applications, without surpassing the threshold of toxicity, the assessment of potential levels of Mg accumulation in tissues, as well as the interactions with Ca and physicochemical properties, prompt this study. An agronomic workflow for Mg enrichment, consisting of six foliar applications of MgSO_4_ with four concentrations (0%, 0.25%, 1% and 4%), equivalent to 0, 43.9, 175.5 and 702 g ha^−1^_,_ was applied on two tomato (*Lycopersicum esculentum* L.) genotypes (Heinz1534 and Heinz9205). During fruit development, leaf gas exchange was screened, with only minor physiological deviations being found. At harvest, Mg contents among tissues and the interactions with Ca were analyzed, and it was found that in both varieties a higher Mg/Ca ratio prevailed in the most external part of the fruit sprayed with 4% MgSO_4_. However, Mg distribution prevailed relatively near the epidermis in H1534, while in H9205 the higher contents of this nutrient occurred in the core of the fruit, which indicated a decrease of the relative proportion of Ca. The morphologic (height and diameter), physical (dry weight and density) and colorimetric parameters, and the total soluble solids of fruits, did not reveal significant changes in both tomato varieties. It was further concluded that foliar application until 4% MgSO_4_ does not have physiological impacts in the fruit’s quality of both varieties, but in spite of the different patterns of Mg accumulation in tissues, if the mean value in the whole fruit is considered, this nutrient prevails in H1534. This study thus suggests that variety H1534 can be used to attain tomato fruits with added value, providing an option of further processing to achieve food products with functional properties, ultimately proving a beneficial option to producers, the food processing industry and consumers. Moreover, the study reinforces the importance of variety choice when designing enrichment workflows.

## 1. Introduction

Magnesium is relatively common in soils, being often available as Mg^2+^, which is a highly mobile chemical form [1,2,3]. Its deficit in soils can be the result of [3,4,5] intensive crop production, few source minerals (dolomite, biotite, chlorite, olivine and magnesite, among others) or because it is easily leached (in part associated with its weaker capacity to bind with negatively charged soil particles, due to its hydrated radius, especially in acidic soils). Ultimately, Mg availability to plants prevails in clayey soils, but its scarcity can lead to low plant availability [4]. Moreover, in soils with a pH higher than 6.5, Mg may become non-exchangeable, whereas at lower pH this nutrient uptake suffers the interference of H^+^ [2]. Dry soils can also limit root growth and compromise nutrient uptakes [2,4], whereas the inadequate use of fertilizers, often focused on the use of NPK fertilizers, may further lead to Mg imbalances in soils [3,4,5], which can aggravate cation competition (namely, with Ca^2+^). In this context, antagonism relations may additionally arise, especially when Mg content in soils is low, since its transporters in higher plants are non-specific [2].

Magnesium fertilization on different crop productions and heterogeneous soil types reveals yield improvements [3], namely, in hybrid tomato [6]. This nutrient plays a central metabolic role in plants, being related to the production of photoassimilates and carbohydrate transport from source-to-link organs [4,5]. Indeed, Mg is considered a crucial part of chlorophylls (i.e., in Mg porphyrins) since it activates many enzymes related to plant growth and stabilizes the nucleic acids [4,7]. Interveinal leaf chlorosis is therefore a deficiency symptom that compromises photosynthesis, plant growth and nutrient uptake [4], prevailing firstly in older leaves and leading to the accumulation of carbohydrates in the leaves (since Mg is linked to both the loading and translocation of these molecules to sink organs) [3,8]. On the other hand, Mg toxicity triggers metabolic disturbances in plants. For instance, it was found that high cytoplasmic concentrations of Mg^2+^ (namely, due to drought conditions) can block chloroplast K^+^ channels in *Spinacia oleracea*, leading to carbon fixation deficits and ROS synthesis [9]. Moreover, Mg is mostly stored in vacuoles if supplied in excess [10].

Calcium (Ca) is an essential ‘secondary’ plant nutrient that has an opposing effect on Mg accumulation as a result of competition for metabolically produced binding compounds [11]. Yet, Ca movement to the fruit is exclusively xylemic and dependent on the abundance of functional vessels [12,13,14] since its its content is greater at the peduncle. Moreover, cellular Ca partitioning and distribution can be a limiting factor, being controlled by complex cellular mechanisms, including the expression and activity of Ca-ATPases, Ca^2+^/H^+^ exchangers (CAXs), and Ca channels [15]. Additionally, Ca supports plant cell walls and serves as an internal biochemical messenger under physical or biochemical stress, stabilizing cellular membranes and therefore ensuring fruit growth and development, as well as fruit quality [16].

Through the application of a workflow for Mg enrichment of fruits from two contrasting hybrid tomato genotypes (varieties Heinz1534-H1534 and Heinz9205-H9205), this study aimed to assess the potential levels of this nutrient accumulation in tissues, the interactions with Ca accumulation and the implications for physicochemical properties.

## 2. Results

The root uptake and translocation of Mg to the shoot might vary with the composition of soils and irrigation water, which eventually can determine the deficiency or toxicity for plants’ physiology. In this context, it was found (Table 1) that the electrical conductivity (EC), pH and organic matter (OM) of the soil from the experimental tomato-growing field were 0.0191 S m ^−1^, 7.1 and 2.61%, respectively. These parameters can be related to mineral’s availability in soils and require higher energy consumption for water uptake by the roots, due to higher salinity or water infiltration and soil quality (Table 1). Regarding mineral element composition (Table 1), Fe showed the highest value (1.27%), followed by K (0.61%), Ca (0.16%), and Mg and P in similar quantities (0.08%). Other minerals were present in smaller quantities (Mn, S, Zn and As, respectively, with 301.0 ppm, 49.1 ppm, 17.1 ppm and 5.63 ppm, with the latest one being toxic).

A slight slope in the SE direction characterizes the geomorphology (Figure 1A–D) of the experimental tomato-growing field, with the northwest part being more elevated. As field relief can have a strong influence on runoff drainage system, the slope was measured to determine its water drainage tendencies, and it was found that (Table 1) 35.6% (more or less 1/3) of the total area is prone to flooding (mainly in southern part), implying less need for irrigation in comparison to the remaining area.

It was further found that the irrigation water used during tomato production (Table 1) is of surface origin (supplied by a dam–GPS coordinates 37°55′47.5″ N, 8°4′52.0″ W), and its chemical facies are of a magnesium calcium bicarbonate type (ionic classification by [17]). The most influential water quality in crop productivity is the salinity hazard (as measured by the EC). A high EC determines a higher inability of plants to compete for water with ions in the soil solution. Moreover, the concentration and composition of ions and the relative proportion of sodium, calcium and magnesium, expressed by the sodium adsorption ratio index (i.e., SAR index) in water, calculated according to the [18], determined its suitability for irrigation [19]. If the water composition is high in sodium and low in calcium and magnesium, the cation-exchange complex, when saturated with sodium, leads to the destruction of soil structure, owing to the dispersion of clay particles. In this context, it was found that EC and SAR revealed values of about 0.0886 S m^−1^ and 1.24, respectively. Accordingly, considering the US Salinity Laboratory Staff [18], in which the EC relates to the salinity hazard, and SAR to sodium hazard, the irrigation water of the experimental tomato-growing field falls in the C3S1 quality category, which represents a high salinity hazard but a low sodium hazard. Moreover, when comparing the pH (7.1) of the irrigation water with the saturation pH (7.8), the Langelier saturation index value is −0.74, indicating water subsaturated in calcium carbonate with moderate corrosive action.

The photosynthetic parameters of tomato plants from both varieties were monitored during foliar spraying with Mg to determine if the threshold of toxicity was reached (i.e., therefore to verify if the synthesis of photoassimilates was impaired). After the 3rd foliar spraying, the net photosynthesis (P_n_), stomatal conductance (g_s_), CO_2_ internal concentration (C_i_), transpiration rate (E) and instantaneous water use efficiency (iWUE) (Table 2) were measured. In H1534, significant differences could not be found for P_n_, g_s_, E and iWUE, but with foliar spraying using MgSO_4_ at 4% a decrease was found for C_i_ (of about 21% relative to the control). Regarding H9205, significant differences could not be found for C_i_ and iWUE, yet treatments with MgSO_4_ at 1% and 4% showed significant decreases of P_n_, g_s_, and E.

The Mg content in leaves after three foliar sprays (Table 3) showed that for variety H1534, although not significantly, Mg-sprayed leaves presented increases of this mineral. A different tendency occurred for variety H9205, with the control presenting the highest value, while all sprayed leaves did not present increases in Mg content.

At harvest, whole fruits of H1534 showed (Table 3), relative to the control, progressively increasing contents of Mg until the highest sprayed concentration, reaching an enrichment index of 18.3% in treatment 4% (with significant differences being found only between this treatment and the control). Moreover, the average values of Mg in the fruits of H9205 did not vary significantly, ranging between ca. 394 and 412 ppm.

Considering the Mg/Ca ratio, it was found (Table 4) that, using as a test system treatment sprayed with MgSO_4_ at 4% in both tomato varieties, a decrease from the peel to the core of the fruit occurred.

In this context, a more detailed analysis of the distribution of Mg in fruits tissues (Table 5) revealed a higher accumulation of Mg in the center of the tomato flesh for variety H1534, while a higher concentration prevailed in the core for H9205.

The height and diameter of fruits of both varieties did not vary significantly (Table 6). The height and diameter of H1534 fruits ranged from 50.0 mm to 54.3 mm and 48.3 mm to 52.0 mm, respectively, whereas H9205 varied (Table 6) from 54.0 mm to 55.7 mm and 45.7 mm to 48.7 mm. In the fruits of H1534, dry weight, density and total soluble solids (Table 6) varied between 6.5–8.0%, 1038 kg m^−3^–1178 kg m^−3^ and 5.0 °Brix–5.9 °Brix, respectively. Yet, only the density of treatment with MgSO_4_ at 4% showed a significantly lower value. The fruits of H9205 showed dry weight, density and total soluble solids values that were not significantly different (Table 6); these parameters ranges were 7.5–8.2%, 999 kg m^−3^–1088 kg m^−3^ and 5.3 °Brix–6.2 °Brix, respectively.

At harvest, the colorimetry (i.e., spectral range varying between 450 and 650 nm) of fruits from both varieties did not reveal significant differences among treatments (Table 7). In both cases, the highest values were identified at 650 nm, which parallels the wavelengths associated with the color red (i.e., corresponding to the color perceived by the human eye). Next, 550 nm and 600 nm showed higher values, in which H9205 appeared to have slightly higher transmittance values. Finally, 450 nm, 500 nm and 570 nm showed the smallest transmittance values, and H9205 appeared to have yet again higher values of 500 nm and 570 nm.

## 3. Discussion

Magnesium (Mg^2+^) is mobile in soils, but it is also highly subjected to leaching [20]. Moreover, competition among cations may also become disadvantageous to Mg^2+^ during a plant’s uptake [11], while productivity and fruit quality are strongly influenced by the morphology of the tomato field as it can affect water availability for plants. Nevertheless, in this workflow for Mg enrichment, heterogeneous interactions with soil morphology were avoided between both varieties as they were planted side-by-side (therefore eliminating different effects of the slight slope on water flow/availability) in the same field (Figure 1; Table 1). Moreover, tomato, namely, varieties H1534 and H9205, can be planted in various soil types, having levels of OM, pH and EC varying between 2–4%, 5.5–7.0, and less than 0.04 S m^−1^, respectively [21], as well as in humid or arid climates [22]. Accordingly, in the tomato experimental field all these parameters fall within this range (Table 1), ensuring nutrient availability for plant uptake. Yet, considering that Mg (0.1–1.0%) and P (0.01–0.1%) contents in soils usually range bellow others such as Ca (0.2–1.5%), K (0.2–3.0%) and Fe (0.5–4.0%) [23], it was found that only Ca (0.16%) and Mg (0.08%) are slightly out of these intervals (Table 1), which, at least for Mg, may be due to leaching.

As the irrigation water has a pH ranging between 6.5–8.4 and the electric conductivity is only slightly higher than 0.07 S m ^−1^ (Table 1), its use for agricultural purposes also does not have any restriction [24]. Additionally, Cl and Na concentrations are lower than 140 mg L^−1^, and to Ca and Mg, respectively. Thus, the SAR index value (1.24) indicates that water has low sodium hazard (allowing its use in most soils), and it implies that there is no toxicity or water percolation in the soil [21,24]. Moreover, the Langelier index does not suggest a tendency to the occurrence of calcium carbonate deposits in the systems used for watering the culture.

Independently of the soil and irrigation water characteristics of the tomato experimental field, considering the concentrations of the foliar spraying with MgSO_4_, through analysis of photosynthetic parameters, the synthesis of photoassimilates was checked during the production cycle to determine if the threshold of toxicity was surpassed. It was found that the Mg accumulation in both genotypes varied differently in leaves (Table 3), which implicated different translocation rates resulting from different metabolic specificity between genotypes. In this context, a comparative analysis between Mg enrichment of tomato leaves from varieties H1534 and H9205 revealed some differences for several photosynthetic parameters (Table 2). Indeed, considering the kinetics of Mg^2+^ mobility in plants [25], the rates of leaf gas exchange of H9205 showed (Table 2), relative to the control, a decrease of net photosynthesis that implicated stomata inhibition with higher doses of MgSO_4_ (1% and 4%), thereby limiting transpiration but without affecting leaf instantaneous water use efficiency. Moreover, H1534 revealed a different pattern (Table 2) as no significant variations of net photosynthesis and stomata conductance were found, which therefore ensured that transpiration and leaf instantaneous water-use efficiency remained unaffected. Moreover, in this variety, the lower internal CO_2_ concentration of treatment 4% suggests higher CO_2_ assimilation, determining the higher values of the net photosynthesis of these treatments (relative to H9205). Nevertheless, during tomato fruit development, foliar spraying with increasing levels of MgSO_4_, without triggering relevant changes to the mobilization of photoassimilates, was coupled with higher total contents of Mg in H1534 but different patterns of Mg accumulation in tissues (Table 3, Table 4 and Table 5). Without reaching toxic levels of Mg accumulation, the higher foliar spraying concentration of MgSO_4_ (4%) promoted (Table 4), for both varieties, a higher Mg/Ca ratio in the most external part of the fruit. However, dividing tomato tissues in five regions, Mg distribution prevailed relatively near the epidermis in H1534, whereas in H9205 the higher contents of this nutrient occurred in the core of the fruit (Table 5), indicating a decrease of the relative proportion of Ca accumulation. The higher contents of Mg (Table 5) in fruit tissues of H9205 (relatively to H1534) further suggest an antagonism of accumulation relatively to Ca. In fact, this trend as long been reported, namely, in soybean, wheat, rice and beans, eventually due to competition for metabolically produced binding compounds [11].

Since varieties H1534 and H9205 are meant for industrial processing, morphology and colorimetry, alongside other quality parameters, gain added importance. Upon arriving at factories, raw foods undergo a selection to maintain the final products characteristics. Thus, for both varieties, maintaining these characteristics or even improving them, despite Mg foliar spraying before harvest, follows a central principle of enrichment, where raw foods must not be affected. At a morphological level, both genotypes have distinct characteristics. In general, independently of Mg spraying, H1534 was slightly smaller in height but slightly larger in diameter, when compared to H9205 (Table 6). Accordingly, both varieties upon Mg enrichment kept their different fruit shapes, with H1534 being classified as “blocky” and H9205 as “oval” [22]. In general, among Mg treatments, dry weight also consistently prevailed higher in H9205 (Table 6), but values were not deviant from what was expected in both varieties. Moreover, density only revealed a significant decrease in treatment with MgSO_4_ at 4% of H1534 (Table 6), but among treatments, H9205 also seemed to be less dense than H1534. This may be because in terms of size, H1534 is classified as “M”, while H9205 is classified as “S”, meaning that H9205 is expected to reach weight values inferior to 70 g, while H1534 should reach values between 70–84 g [22]. Dry weight values of variety H1534 were also in accordance with other studies [26].

Of all the studied parameters, total soluble solids and color have the highest relevance [27]. Since tomato flavor can result from a balance of sweet and sourness and Mg is responsible for metabolite synthesis as well as its translocation to fruits, an effect on flavor characteristics can happen [25]. Thus, regarding total soluble solids content (Table 6), it is important to observe that no significant differences were identified in both varieties, and in both cases all values were equal or higher than five, a trait often appreciated for industrial purposes and observed by another study with the same variety (H1534) [26]. Regarding color (Table 7), both varieties evidenced a higher transmittance in the wavelength corresponding to red, thus reaching a high lycopene content, which is a major source for human diets [27]. Lycopene is a pigment linked with a ripening of tomato fruits, resulting from carotenoids synthesis and chlorophyll degradation, with powerful antioxidant activity. Because lycopene is rather stable when processed, it can be used to make the most processed, enriched tomato products for consumers [28]. Since, in general, morphological, colorimetric and quality parameters did not reveal significant differences among the enrichment treatments, it was found that Mg concentrations sprayed in the pre-harvest phase did not compromise its acceptability by the agro-industries.

## 4. Materials and Methods

### 4.1. Agricultural Field and Workflow for Mg Enrichment

The experimental tomato-growing field, with 990 m^2^, was located in the center-south of Portugal (according to European Datum-GPS coordinates 37°56′55.360″ N; 8°10′26.092″ W). Two industrial varieties of tomatoes (Heinz1534-H1534 and Heinz9205-H9205) were selected for Mg enrichment, being planted side by side in a brown Mediterranean soil, of sandy clay loam texture, mapped as Luvisol [29] to avoid a heterogeneous effect of soil and morphology and different water flow influence during irrigation. Two tomato varieties (*Lycopersicum esculentum* L.) were planted on the 10th of May and harvested on the 5th of September of 2018. Sections with 9 m · 1 m (compass of 1.50 m · 0.20 m, with 54 plants each) were used for both varieties. The technical itinerary for tomato enrichment with Mg consisted of six foliar applications, spaced seven days between each other. The foliar spray was administered in four concentrations (0%; 0.25%, 1% and 4%) of MgSO_4_, equivalent to 0, 43.9, 175.5 and 702 g ha^−1^, respectively. Four replicates per concentration (corresponding to 216 plants), with a total of 864 plants, were planted for each variety. During the agricultural period, air temperatures reached (Figure 2) a daily average of 30/14 °C (maximum/minimum, varying between 6/44 °C). Total precipitation during the production cycle (118 days) was ca. 50.5 mm (mean value of 0.43 mm day^−1^), with daily maximum and minimum values of 19.05 and 0 mm, respectively.

### 4.2. Orthophotomaping

Orthophotomaps of the experimental field of the growing tomato varieties were produced using a high-definition and multi-sector RGB camera (with three electromagnetic spectra bands–Red, Green and Blue) and a parrot sequoia camera (with five electromagnetic spectra bands – NIR, REG, green, red and RGB) installed in a drone. The calibration of the camera (parrot sequoia) took into account the environmental brightness conditions. The images were processed with Workstation (AORUS, GIGA-BYTE Technology Co., Ltd.-2019). The drainage patterns of the surface water and the geomorphology of the field were studied with an Agisoft PhotoScan Professional (Version 1.2.6, Software of 2016 and the ESRI of 2011 and ArcGIS Desktop-Release 10 from Redlands, CA: Environmental Systems Research Institute). The classification of surface water drainage areas followed [30]. The lower class corresponds to flattened surfaces, more prompted for the accumulation of surface water, representing potential infiltration areas. The highest class represents zones that, due to their morphology, promote surface-water runoff, having a reduced amount of water infiltration.

### 4.3. Soil and Irrigation Water Analysis

A hexagonal grid (4.50 m × 6.60 m) was applied in the experimental field of the growing tomato varieties, and 16 soil samples (100 g, picked up at 30 cm depth) were collected for physical and chemical analysis. Soil samples were passed through a 2 mm nylon sieve to remove major debris before analysis. After drying at 105 °C for 24 h until constant weight was achieved, soil moisture was determined. Organic matter was then estimated after combustion for 4 h, at 550 °C. Following [31], the quantification of minerals was carried out by X-ray fluorescence, using a Thermo Scientific Niton XL3t 950 He GOLDD+ XRF. Electrical conductivity and pH were measured with a multiparameter analyzer (Consort C6030), in a soil mixture with water (1:2.5 g _soil_ mL^−1^ _water milli-q_), under stirring, after a thermal bath (25 °C) for 30 min.

The chemical parameters of the irrigation water were determined by ionic chromatography (Mg, Ca, Na and K), using a Metrohm, 761 Compact IC, equipped with a column and pre-column (Metrosep cation 1-2, 6.1010.000). The eluent mixture (4 mM tartaric acid/1 mM dipicolinic acid) at a flow rate of 1.00 mL/min was used with a sample injection of 10.0 μL [32]. Titration (HCO_3_) in 100 mL of water samples was carried out using 0.1 N hydrochloric acid as titrant, in the presence of 0.1% methyl orange. Photometry (Cl, SO_4_ and NO_3_) was carried out using a Spectroquant NOVA 60, Merck, and specific kits (1.14897, 1.14779, 1.14773 and 1.14842). The multiparameter analyzer Consort C6030, coupled with SP21 and SK20T electrodes, was used to measure the pH, EC and temperature of the water. These parameters were considered to classify the water in terms of dominant ions [17], irrigation purposes (Wilcox Diagram [19]) and index SAR (sodium adsorption ratio) and its corrosive or incrusting action (Langelier saturation index and equilibrium pH).

### 4.4. Leaf Gas Exchange Measurements

Leaf gas exchange parameters were determined according to [33], in 4 to 6 randomized leaves per treatment, on the 24th of July 2018. Net photosynthesis rate (P_n_), stomatal conductance (g_s_), internal CO_2_ concentration (C_i_) and transpiration rate (E) were measured under photosynthetic steady-state conditions after 2 h of illumination (corresponding to middle of the morning). A portable open-system infrared gas analyzer (Li-Cor 6400, LiCor, Lincoln, NE, USA) was used under environmental conditions, with PPFD ranging between 1200 and 1400 µmol m^−2^ s^−1^ and external CO_2_ of *ca*. 400 ppm.

The ratio of P_n_-to-E (which depicts the units of assimilated CO_2_ per unit of water lost through transpiration) allowed for the calculation of leaf instantaneous water-use efficiency (iWUE).

### 4.5. Magnesium Content in Leaves and Tissue Localization

According to [34], Mg content in leaves was assessed with an X-Ray fluorescence analyzer after three foliar sprays (on 24 July 2018). A semi-quantitative distribution of Mg in tissues was carried out by scanning electron microscopy coupled with energy dispersive X-rays spectroscopy, according to [35]. A JEOL *JSMT330A* model coupled with an energy-dispersive X-ray spectroscopy (EDS) and a Tracor Northern Series II microanalyzer was used to assess the rates of Mg/Ca in fruits from the control and the highest concentration of foliar spraying (MgSO_4_ at 4%), considering two different transverse tomato zones from the interior (1) to the exterior (2). Fruits were cut longitudinally, dehydrated and dried in CO_2_, using a Balzers Union CPD 020 system. Then, samples were adhered to 13 mm aluminum stubs with conductive carbon adhesive pads and sputter-coated with a 60/40 Au/Pd alloy, to an approximate thickness of 10 to 15 nm, using a Polaron equipment coating unit. Sputter-coated were examined, imaged and analyzed at 20 Kv, using a static electron beam that interacted with the sample producing a variety of emissions, including characteristic X-rays of different elements. The abundance of specific elements was determined, in 200 s and after 5000 readings.

Following [36], a quantitative distribution of Mg in tissues was further characterized, using a µ-EDXRF system (M4 Tornado™, Bruker, Germany), in fruits from the control and from the highest concentration of foliar spraying (MgSO_4_ at 4%), after defining five zones from the exterior (1) to the interior (5). Fruits were cut at the equatorial section into slices, with a stainless-steel surgical blade. The X-ray generator was operated at 50 kV and 100 µA, without the use of filters, to enhance the ionization of low-Z elements. To achieve better quantification of Mg, a set of filters between the X-ray tube and the sample, composed of three foils of Al/Ti/Cu (with a thickness of 100/50/25 µm, respectively), was used. All the measurements with filters were performed with 600 µA current. The detection of fluorescence radiation was performed by an energy-dispersive silicon drift detector, XFlash™, with a 30 mm^2^ sensitive area and an energy resolution of 142 eV for Mn Kα. Measurements were carried out under 20 mbar vacuum conditions. These point spectra were acquired during 200 s.

### 4.6. Morphometric, Colorimetric and Total Soluble Solids of Fruits

Height, diameter, density and dry weight were measured in four tomato fruits from each treatment. In the epidermis of fruits, the colorimetric parameters were measured, using a scanning spectrophotometric colorimeter (Agrosta, European Union), equipped with a sensor having a half-max width of 40 nm (covering the visible region of the electromagnetic spectrum) and 6 phototransistors with sensibility in a specific region of the spectrum (380 nm—violet; 450 nm—blue; 500 nm—green; 570 nm—yellow; 600 nm—orange; and 670 nm—red) [37]. Light was provided by a white LED covering all the visible regions. Using a digital refractometer Atago (Tokyo, Japan), total soluble solids, expressed in °Brix, were measured.

### 4.7. Statistical Analysis

A statistical analysis of the results was performed using a one-way ANOVA (*p* ≤ 0.05) to assess the differences between all treatments in different parameters for each variety. Additionally, a Tukey’s was applied for mean comparison (with a confidence level of 95%).

## 5. Conclusions

Magnesium is an important nutrient for plants, often less regarded for their health and nutrition management. In fact, it is an important co-factor of several enzymes, namely, those involved in photosynthetic functioning, and it integrates the structure of the chlorophyll molecules. Therefore, Mg largely determines the ratio of photoassimilates synthesis and mobilization through the source-sink metabolism. In this context, it was found that Mg accumulation of tomato fruits through foliar spraying with MgSO_4_ (4%, equivalent to 702 g ha^−1^) in the pre-harvest phase did not become toxic. Moreover, considering that genotypes H1534 and H9205 have different characteristics for Mg mobilization and accumulation in tissues, it is further concluded that the choice of the tomato genotype is of crucial importance. Obtaining tomato fruits with added value throught Mg enrichment, as well as considered its natural compounds (like lycopene), and versatility of products to be obtained (that go along with healthy food choices), indicates that working with this culture can reveal itself to be an opportunity for agro-industries. Yet, for this purpose, not only the workflow for fertilization must be considered, but the genotype specificity also needs to be determined.

## Figures and Tables

**Figure 1 plants-11-01854-f001:**
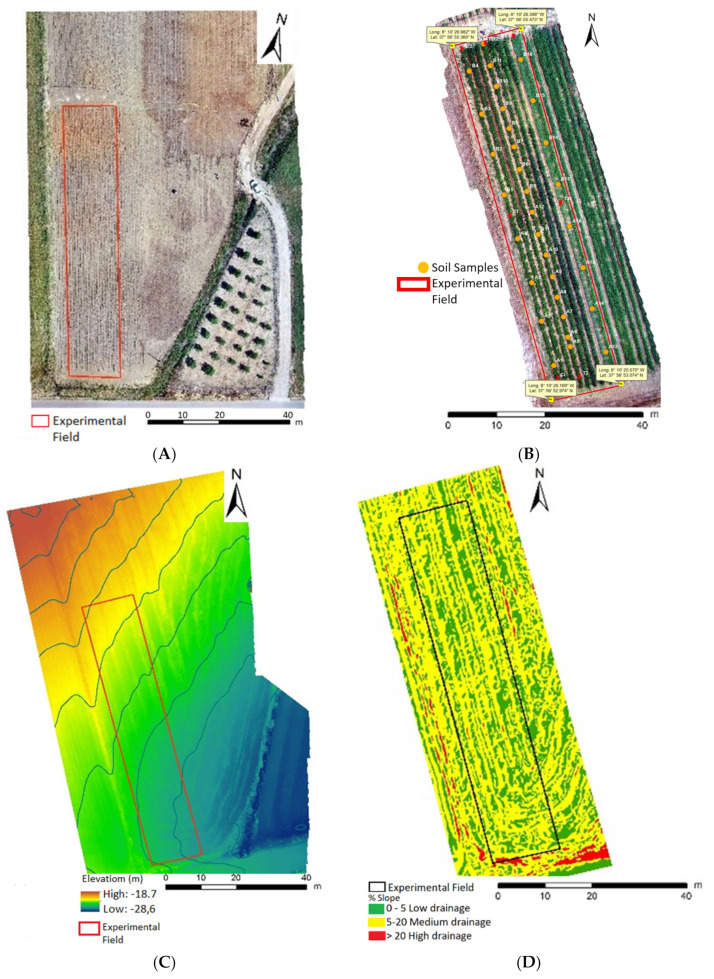
Orthophotomaps from the experimental tomato-growing field of *Lycopersicum esculentum* L., varieties H1534 and H9205. (**A**) Indication of the parcel’s limits; (**B**) sampling points of soil analysis; (**C**) field digital elevation model; and (**D**) slopes digital map.

**Figure 2 plants-11-01854-f002:**
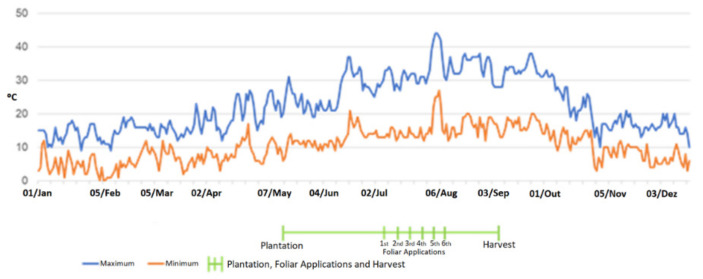
Maximum and minimum daily temperatures during tomato (*Lycopersicum esculentum* L., varieties H1534 and H9205) production cycle and dates of foliar spraying with MgSO_4_.

**Table 1 plants-11-01854-t001:** Physical and chemical parameters of the soil and irrigation water from the experimental tomato-growing field (of H1534 and H9205 varieties), selected for fruits enrichment with Mg.

Physical and Chemical Parameters of Soil Collected from 0–30 cm of Depth											
pH	Electrical Conductivity	Organic Matter	Fe	K	Ca	Mg	P	Mn	S	Zn	As
	S m ^−1^	%	%					ppm			
7.1	0.0191	2.61	1.27	0.61	0.16	0.08	0.08	301.0	49.1	17.1	5.63
**Physical and Chemical Parameters of Irrigation Water**											
**pH**	**Electrical Conductivity**	**Ca^2+^**	**K^+^**	**Mg^2+^**	**Na^+^**	**Cl^−^**	**HCO_3_^−^**	**SO_4_^2−^**		**NO_3_^−^**	**PO_4_^3−^**
	S m ^−1^	mg L^−1^ (meq L^−1^)									
7.1	0.0886	58.9 (2.9)	5.9 (0.1)	34.7 (2.8)	48.6 (2.1)	77.1 (2.2)	225.0 (3.6)	78.0 (1.6)		0.2 (0.004)	<1.5 (<0.04)
**Area of Agricultural Parcel Under Effect of High or Low Water Accumulation**											
**Slope Classes (%)**		**Drainage Type**			**Partial Area (m^2^)**			**Partial Area (%)**			
1− [0–5%]		Low			391.7			35.6			
2− [5–20%]		Medium			705.5			64.0			
3− >20%		High			4.5			0.41			

**Table 2 plants-11-01854-t002:** Net photosynthesis (P_n_), stomatal conductance to water vapor (g_s_), CO_2_ internal concentration (C_i_) and transpiration (E) rates, as well as variation in the instantaneous water use efficiency (iWUE) in leaves of *Lycopersicum esculentum* L., varieties H1534 and H9205, submitted to Mg enrichment after the 3rd foliar spraying with Mg. Each value is the mean value ± SE (*n* = 4). Different letters indicate significant differences between treatments for the same analytical date (a, b) within each genotype (statistical analysis using the single factor ANOVA test, *p* ≤ 0.05). Foliar spray was carried out with 3 concentrations (0.25%, 1% and 4%) of MgSO_4_, equivalent to 43.9, 175.5 and 702 g ha^−1^, respectively. Control (Ctr) is equal to 0%.

Treatments	Date of Measured Parameters, 24 July 2018							
	H1534				H9205			
**P_n_ (µmol CO_2_ m^−2^ s^−1^)**								
**Ctr**	15.26	±	1.44	a	18.51	±	1.56	a
**0.25%**	13.08	±	1.45	a	14.60	±	0.67	ab
**1%**	15.16	±	1.50	a	11.60	±	1.37	b
**4%**	13.72	±	1.25	a	10.05	±	1.62	b
**g_s_ (mmol H_2_O m^−2^ s^−1^)**								
**Ctr**	264.6	±	21.9	a	300.5	±	28.3	a
**0.25%**	298.6	±	9.2	a	309.5	±	36.7	a
**1%**	241.7	±	19.7	a	191.1	±	23.1	b
**4%**	241.5	±	42.4	a	169.7	±	35.1	b
**Ci (ppm)**								
**Ctr**	199.7	±	5.9	ab	186.4	±	8.2	a
**0.25%**	211.2	±	7.4	a	183.1	±	6.4	a
**1%**	196.8	±	2.0	ab	184.8	±	3.6	a
**4%**	158.4	±	24.0	b	193.6	±	2.8	a
**E (mmol H_2_O m^−2^ s^−1^)**								
**Ctr**	4.20	±	0.24	a	4.63	±	0.23	a
**0.25%**	4.25	±	0.18	a	3.97	±	0.14	ab
**1%**	4.00	±	0.13	a	3.57	±	0.18	b
**4%**	3.76	±	0.21	a	3.36	±	0.16	b
**iWUE (mmol CO_2_ mol^−1^ H_2_O)**								
**Ctr**	3.58	±	0.19	a	3.94	±	0.21	a
**0.25%**	3.00	±	0.24	a	3.78	±	0.28	a
**1%**	3.76	±	0.34	a	3.23	±	0.29	a
**4%**	3.85	±	0.51	a	3.90	±	0.31	a

**Table 3 plants-11-01854-t003:** Mean values of Mg contents ± SE (*n* = 3) in leaves and whole the tomato fruits of *Lycopersicum esculentum* L., varieties H1534 and H9205, after the 3rd foliar spraying and at harvest, respectively. Letters a and b indicate significant differences, of each parameter, between the treatments (statistical analysis using the single factor ANOVA test, *p* ≤ 0.05). Foliar spray was carried out with 3 concentrations (0.25%, 1% and 4%) of MgSO_4_, equivalent to 43.9, 175.5 and 702 g ha^−1^, respectively. Control (Ctr) is equal to 0%.

Treatments	Mg Contents in Leaves (% ± SE)				Mg Contents in Whole Fruits (ppm ± SE)			
	Variety H1534		Variety H9205		Variety H1534		Variety H9205	
**Crt**	0.31 a	±0.05	0.39 a	±0.00	404 b	±9	412 a	±18
**0.25%**	0.37 a	±0.08	0.33 a	±0.00	435 ab	±0	412 a	±12
**1%**	0.39 a	±0.09	0.34 a	±0.01	437 ab	±10	399 a	±3
**4%**	0.39 a	±0.03	0.35 a	±0.03	478 a	±12	394 a	±7

**Table 4 plants-11-01854-t004:** X-rays dispersive spectroscopy (SEM-EDS) of fruit samples (*n* = 4) of *Lycopersicum esculentum* L., varieties H1534 and H9205, with indication of 2 transverse sections (1—central zone, and 2—peripheral zone next to the epidermis), where the readings were taken, to calculate the ratio between the Mg/Ca ratio of treatment 4% (immediately after harvest).

Location of the 2 Transverse Sections in the Longitudinal Section of Tomato Fruit	Ratio of Mg/Ca in the Transverse Section of the Fruit
Mg/Ca	Zones
**H1534**
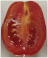	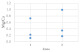	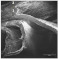
**H9205**
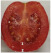	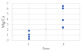	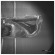

**Table 5 plants-11-01854-t005:** Fruit segments and average of Mg contents ± SE (*n* = 3) of five sections selected in the equatorial region of tomato fruits of *Lycopersicum esculentum* L., varieties H1534 and H9205, ranging from epidermis (1) to the locular cavity (5) of the control and 4% treatment. Foliar spray was carried out with 3 concentrations (0.25%, 1% and 4%) of MgSO_4_, equivalent to 0, 43.9, 175.5 and 702 g ha^−1^. Control (Ctr) is equal to 0%.

Fruit Segments in the Transverse Section	Macroscopic Visualization of Fruit Segments in the Longitudinal Sections	Fruit Segments in the Transverse Section
**H1534**
**0%**		**4%**
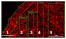	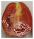	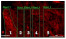
**Average values of Mg contents ± SE of each section**		**Average values of Mg contents ± SE of each section**
0.74 ± 0.04	0.00 ± 0.00	0.45 ± 0.00	0.00 ± 0.00	0.00 ± 0.00	0.10 ± 0.00	2.73 ± 0.14	2.78 ± 0.14	0.00 ± 0.00	1.00 ± 0.05
**H9205**
**0%**		**4%**
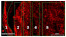	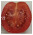	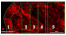
**Average values of Mg contents ± SE of each section**		**Average values of Mg contents ± SE of each section**
0.00 ± 0.00	0.00 ± 0.00	1.33 ± 0.07	0.00 ± 0.00	4.76 ± 0.24	2.24 ± 0.11	4.63 ± 0.23	5.97 ± 0.30	13.1 ± 0.66	9.72 ± 0.49

**Table 6 plants-11-01854-t006:** Height, diameter, dry weight, density and total soluble solids of fruits from *Lycopersicum esculentum* L., varieties H1534 and H9205, at harvest. Letters a and b indicate significant differences of each parameter, between the treatments (statistical analysis using the single factor ANOVA test, *p* ≤ 0.05). Each value is the mean ± SE (*n* = 3). Foliar spray was carried out with 3 concentrations (0.25%, 1% and 4%) of MgSO_4_, equivalent to 43.9, 175.5 and 702 g ha^−1^, respectively. Control (Ctr) is equal to 0%.

Treatments	Height (mm)		Diameter (mm)		Dry Weight (%)		Density (kg m^−3^)		°Brix	
	Mean	SE	Mean	SE	Mean	SE	Mean	SE	Mean	SE
	**Tomato Variety H1534**									
**Crt**	54.3 a	±0.9	48.7 a	±1.7	6.5 a	±0.3	1151 a	±19	5.0 a	±0.0
**0.25%**	51.3 a	±0.9	52.0 a	±0.8	7.0 a	±0.3	1178 a	±43	5.9 a	±0.1
**1%**	50.0 a	±2.1	48.3 a	±2.5	8.0 a	±0.4	1106 ab	±6	5.1 a	±0.1
**4%**	50.0 a	±1.5	50.3 a	±1.2	7.3 a	±0.3	1038 b	±12	5.3 a	±0.2
	**Tomato Variety H9205**									
**Crt**	55.3 a	±3.3	48.7 a	±1.3	7.5 a	±0.2	1052 a	±22	6.2 a	±0.1
**0.25%**	54.0 a	±3.5	48.3 a	±1.3	8.2 a	±0.0	1042 a	±3	5.3 a	±0.3
**1%**	55.0 a	±1.0	45.7 a	±1.8	8.0 a	±0.2	999 a	±69	5.7 a	±0.3
**4%**	55.7 a	±2.2	47.0 a	±0.6	7.7 a	±0.3	1088 a	±25	5.4 a	±0.2

**Table 7 plants-11-01854-t007:** Colorimetric analysis along the visible region of the electromagnetic spectrum of fruits from *Lycopersicum esculentum* L., varieties H1534 and H9205, at harvest. Letter a indicates the absence of significant differences, of each parameter, between treatments (statistical analysis using the single factor ANOVA test, *p* ≤ 0.05). Foliar spray was carried out with 3 concentrations (0.25%, 1% and 4%) of MgSO_4_, equivalent to 43.9, 175.5 and 702 g ha^−1^, respectively. Control (Ctr) is equal to 0%.

Treatments	Wavelength (nm)											
	450		500		550		570		600		650	
	Mean	SE	Mean	SE	Mean	SE	Mean	SE	Mean	SE	Mean	SE
	**Tomato Variety H1534**											
**Crt**	567 a	±4	452 a	±5	706 a	±6	398 a	±7	659 a	±6	1433 a	±9
**0.25%**	570 a	±5	454 a	±5	705 a	±3	399 a	±5	660 a	±12	1412 a	±14
**1%**	564 a	±2	447 a	±2	701 a	±3	398 a	±6	674 a	±13	1425 a	±30
**4%**	560 a	±5	444 a	±6	696 a	±5	395 a	±7	675 a	±9	1447 a	±6
	**Tomato Variety H9205**											
**Crt**	574 a	±7	460 a	±5	717 a	±8,	415 a	±15	704 a	±36	1451 a	±30
**0.25%**	577 a	±8	466 a	±11	734 a	±16	419 a	±15	699 a	±23	1462 a	±10
**1%**	567 a	±6	457 a	±7	719 a	±17	411 a	±16	703 a	±30	1483 a	±20
**4%**	573 a	±2	460 a	±3	721 a	±6	419 a	±5	714 a	±13	1491 a	±14

## Data Availability

Not applicable.

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
