# Peer review of "Magnesium Accumulation in Two Contrasting Varieties of Lycopersicum esculentum L. Fruits: Interaction with Calcium at Tissue Level and Implications on Quality"

_plants, 2022, doi:10.3390/plants11141854_

Round 1
Reviewer 1 Report
1. The manuscript is poorly written and the language needs to be improved.
2. Time units used in the materials and methods should be consistent.
3. What’s the basis for dividing tomato fruit into five zones and what’s the point of it?
4. It’s obvious that Magnesium hasn’t been enriched in the fruits of Variety H9205 after foliar spray of Mg. Therefore, the title seemed inappropriate.
5. There lacks connections and logic among different parts (photosynthesis, Mg content of fruits and the fruit quality).
6. What’s the effect of foliar application of Mg on the Mg content in leaves. Is it affected or not? And how does leaf Mg content affect fruit Mg content.
Reviewer 2 Report
The paper is good, however the reader my launch some questions and remarks.
First of all, the article is too long and too broad for such purpose. It is more of a comprehensive study on the performance of two tomato varieties than a regular scientific paper.
The value of the work is in its broad spectrum of scientific observations and measurements. However the weakness is that the conclusion is just the presentation of a few statements concerning the varieties studied.
I suggest the authors to present a target-oriented discussion regarding the various observations, because the borderless description of the results may lead to confusion only. Hints: the less would have been more in this case.
Finally a good summary should involve the opinion of the authors concerning the case.
Round 2
Reviewer 1 Report
1. What’s the basis for dividing tomato fruit into five zones and what’s the point of it? It seems meaningless.
2. No significant difference in the Mg concentration of leaves was observed among control and Mg application treatments. Therefore, is it possible that Mg applied to the leaves was not absorbed into the plant, which means the treatment was meaningless. If not so, what’s the reasonable reason?
3. It’s obvious that the accumulation of Mg in fruits was increased for Variety H1534 but decreased for Variety H9205 after the foliar spray of Mg, even though not significantly. So what is the mechanism difference between the two varieties in response to the application of Mg?
4. Lin 77: use “enzymes related to plant growth” instead of “plants growth related enzymes”
5. Line 164-166, 174, 261, 263 and Table 3: use leaves instead of leaf’s or leafs
6. Line 193-194: It can be changed to “revealed a higher accumulation of Mg in the center of the tomato flesh for variety H1534, while a higher concentration prevailed in the core for H9205”.
7. Line 324: use “on the 5th of September” instead of “in the 5th of September”; and the same goes for Line 377
8. Line 437: use “that the choice of the tomato genotype is of crucial importance” instead of “that it is of crucial importance is the choice of the tomato genotype”
Reviewer 2 Report
I have received the improved version of the authors. There are positive signs concerning the improvement in the topic regarding the research topics. Negative is still the abundance of information - the "logorrhea".
I suggest the authors to précise and shorten the material. With a better reshaping I support the edition of the paper.
Round 3
Reviewer 1 Report
The authors addressed all my comments.